# CircRNA-PI4KB Induces Hepatic Lipid Deposition in Non-Alcoholic Fatty Liver Disease by Transporting miRNA-122 to Extra-Hepatocytes

**DOI:** 10.3390/ijms24021297

**Published:** 2023-01-09

**Authors:** Chang-Hai Liu, Wei Jiang, Qingmin Zeng, Dongbo Wu, Hong Li, Lingyun Zhou, Lang Bai, Hong Tang

**Affiliations:** 1Center of Infectious Diseases, West China Hospital, Sichuan University, Chengdu 610041, China; 2Division of Infectious Diseases, State Key Laboratory of Biotherapy and Center of Infectious Disease, West China Hospital, Sichuan University, Chengdu 610041, China

**Keywords:** non-alcoholic fatty liver disease, circRNA, miRNA, lipid deposition

## Abstract

Ectopic fat deposition in the liver, known as non-alcoholic fatty liver disease (NAFLD), affects up to 30% of the worldwide population. miRNA-122, the most abundant liver-specific miRNA, protects hepatic steatosis and inhibits cholesterol and fatty acid synthesis in NAFLD. Previously, we have shown that compared with its expression in healthy controls, miRNA-122 decreased in the liver tissue but gradually increased in the serum of patients with non-alcoholic fatty liver disease and non-alcoholic steatohepatitis, suggesting that miRNA-122 could have been transported to the serum. Here, we aimed to confirm and unravel the mechanism of transportation of miRNA-122 to extra-hepatocytes. Our findings showed a decrease in the intra-hepatocyte miRNA-122 and an increase in the extra-hepatocyte (medium level) miRNA-122, suggesting the miRNA-122 “escaped” from the intra-hepatocyte due to an increased extra-hepatocyte excretion. Using bioinformatics tools, we showed that miRNA-122 binds to circPI4KB, which was further validated by an RNA pull-down and luciferase reporter assay. The levels of circPI4KB in intra- and extra-hepatocytes corresponded to that of miRNA-122, and the overexpression of circPI4KB increased the miRNA-122 in extra-hepatocytes, consequently accomplishing a decreased protective role of miRNA-122 in inhibiting the lipid deposition. The present study provides a new explanation for the pathogenesis of the hepatic lipid deposition in NAFLD.

## 1. Introduction

Ectopic fat deposition in the liver, known as non-alcoholic fatty liver disease (NAFLD), affects up to 30% of the worldwide population [1,2,3]. NAFLD encompasses a wide spectrum of liver damage, ranging from non-alcoholic fatty liver (NAFL) to non-alcoholic steatohepatitis (NASH). NAFL is defined as the presence of hepatocyte steatosis without evidence of inflammation. NAFL is an independent predictor for insulin resistance and cardiovascular risk [4,5,6,7], while NASH is often more progressive, eventually advancing to cirrhosis and hepatocellular carcinoma (HCC) [2,3].

MicroRNAs (miRNA) are non-protein-coding, small single-stranded RNA, typically 21 to 23 nucleotides long, that regulate the gene expression via messenger RNA (mRNA) degradation or translational inhibition [8,9]. Several miRNAs, including miR-34a and miRNA-21, showed important roles in the control of the hepatic lipid metabolic pathways by targeting key transcription factors, including the SIRT1 activity and beta-oxidation genes [10]. miR-192 is one of several obesity-associated exosomal miRNAs; its expression in the circulation is elevated in both simple steatosis and NASH compared with the healthy controls while its liver expression is elevated in simple steatosis but in NASH [9]. miRNA-122 is the most abundant liver-specific miRNA (250,000 copies/hepatocyte), accounting for approximately 70% of the total liver miRNAs [11]. miRNA-122 has been shown to protect hepatic steatosis in two ways: (1) it inhibits the cholesterol synthesis by post-transcriptionally targeting the mRNA level of fatty acid synthase (*FAS*), 3-hydroxy-3-methyl-glutaryl coenzyme-A reductase (*HMG-CoA* reductase, *HMGCR*), and their transcriptional activators, the sterol-response-element binding protein 2 (*SREBP-2*) [12]. (2) It reduces the fatty acid synthesis by post-transcriptionally targeting the mRNA level of 1-acyl-sn-glycerol-3-phosphate acyltransferase (*AGPAT1*) and Diacylglycerol O-acyltransferase 1 (*DGAT1*) [13]. Furthermore, miRNA-122 has been demonstrated to be upregulated in every scenario of NAFL, NASH, and fibrosis, rendering it as a potential miRNA-based serum biomarker [14]. Our study also showed that several miRNAs had an inconsistent or inverse correlation between the circulating and liver tissue expression [14]. In particular, the miRNA-122 decreased in the liver tissues and gradually increased in the serum of patients with NAFL and NASH compared to its levels in the healthy controls [14]. The study suggested the possible relocation of miRNA-122 from the hepatocyte to serum; however, the underlying mechanism of the relocation of miRNA-122 from the hepatocyte to serum is not clear and, therefore, needs further studies.

Covalently closed circRNAs were discovered nearly 40 years ago and were often considered as “background noise” of aberrant splicing byproducts with little functional potential [15]. However, more and more studies have proved that circRNA plays an important role in physiological or pathological conditions. Several studies have described the role of circRNAs in the pathogenesis of NAFLD. For example, circRNA_0046367 and circRNA_0046366, two homologous circRNA transcripts from *FASN*, ameliorated the oxidative stress based on a steatosis attenuation by sponging miR-34a, thus regulating the PPARα regulatory system [16]. Moreover, the circHIPK3/miR-192-5p/forkhead box O1 signaling pathway has been shown to be involved in the development of adipogenesis, IR, and hepatic steatosis [17]. Most studies suggest that circRNAs can bind to miRNAs to regulate the biological function of downstream proteins. circScd1 was expressed at low levels in NAFLD tissues, and the aberrant expression of circScd1 affects the extent of hepatocellular lipidosis in NAFLD and promotes steatosis via the JAK2/STAT5 pathway [18]. The knockdown of circH19 drives an adipogenic genes expression, accompanied by a lipid accumulation by triggering the sterol-regulatory SREBP1 translocation from the cytoplasm to the nucleus through an interaction with a polypyrimidine tract-binding protein 1 [19]. In addition, miRNAs have also been implicated in obesity. The knockdown of circSAMD4A in the adipose tissues of obese mice increased the insulin sensitivity, glucose tolerance, and energy expenditure through the miR-138-5p/EZH2 axis [20]. The mitochondria-located circRNA SCAR in NASH fibroblasts was shown to directly bind to the ATP5B of the ATP synthase in a mitochondrial permeability transition pore complex and blocks the cyclophilin D-mPTP interaction, inhibiting the ROS generation, mROS output, and fibroblast activation, thus activating steatosis-to-NASH progression [21]. It has been shown that the circRNA_002581–miR-122–CPEB1 axis actively participates in the pathogenesis of NASH through the PTEN–AMPK–mTOR pathway-related autophagy suppression [22].

More recently, it has been reported that circRNAs could regulate the stability or transport of miRNA in neurons [23]. Li et al. reported that the overexpression of miR-7 into the cells decreased the CDR1as circRNA level in the exosomes but slightly increased it in both HEK293T and MCF-7 cells [24]. However, no studies to date have described whether the overexpression of circRNA could influence the distribution of the intra- and extra-cell miRNA. Therefore, we hypothesized that the hepatic protective miRNA-122 was carried from intra- to extra-hepatocytes by circRNA, resulting in a decreased level of miRNA-122 in intra-hepatocytes and thus the appearance of lipid deposition. In addition, we also speculate that due to the enhancement of the stability of the miRNA-122 by binding to the circRNA, the serum levels of the miRNA-122 gradually increased according to the severity of the NAFLD.

## 2. Results

### 2.1. The Decreased Level of miRNA-122 in Hepatocytes Is Due to the Increased Excretion of miRNA-122 to Extra-Hepatocytes

We successfully established the in vitro model of NAFLD by culturing with oleic and palmitic acids at a final concentration of 0.5 mM (FFA) for 24 h, and the lipid deposition was assessed by oil red O staining (Figure 1A). We used this model to investigate the possible causes of the upregulation of the extracellular miRNA-122. The pre-miRNAs in the nucleus are usually exported to the cytoplasm and cleaved to mature miRNAs; therefore, the pre-miRNAs serve as a synthesis indicator of the miRNAs. The PCR results showed that the intra-hepatocyte pre-miRNA-122 level was increased after FFA-induced steatosis, however, the level of intra-hepatocyte miRNA-122 decreased, and that of the extra-hepatocyte (medium level) miRNA-122 increased (Figure 1B). These findings suggest that the miRNA-122 “escapes” from the hepatocytes due to the increased excretion to extra-hepatocytes. To further assess the miR-122 in an in vivo model, male C57BL/6 mice aged 6–8 weeks were fed a Western diet (WD) and drinking water containing high glucose for 12 weeks to establish an NAFLD mouse model. Oil red O staining, H&E staining, and micro-CT (relative quantitative measurement of abdomen adipose tissue) were used to assess the hepatic steatosis, ballooning, and abdomen adipose tissue (Figure 1C). The level of miRNA-122 in the liver tissues of the NAFLD mice was decreased, wherein it was increased in the serum compared to the control mice (Figure 1D).

### 2.2. The Predicted miRNA-122-Binding circRNA-PI4KB Showed a Decreased Level in Hepatocytes and an Increased Level in Extra-Hepatocytes

Previous studies have demonstrated that circRNAs regulate the miRNA stability or transport miRNA; therefore, we intended to find the circRNAs specifically sponging to miRNA-122 and transporting miRNA-122 to extra-hepatocyte. The first aim was to predict all circRNAs that could sponge to miRNA-122 in multiple online software. Thus, the ENCORI, StarBase, and Circbank databases predicted 858 circRNAs that could sponge to miRNA-122 as the first dataset (with three databases intersected by the Venn map) (Figure 2A). Subsequently, the intersection of these predicted circRNAs with liver-specific circRNAs in the CircBase database and GSE134146 dataset identified 37 circRNAs, which were selected for a further validation in the NAFLD cell model. RT-qPCR revealed that the expression levels of 5 of these 37 circRNAs, including circC7orf44, circSPECC1-2, circPI4KB, circRBBP8, and circAFF1-1, differed significantly in the FFA-induced NAFLD cell model compared to the control cells (Figure 2B). Furthermore, circPI4KB decreased in L02 cells and increased in an FFA-induced NAFLD culture medium (Figure 2C), with the same tendency as miRNA-122, which was previously shown in Figure 1B. To confirm these results in the mouse model, we evaluated the level of circPI4KB in the liver tissues and the serum of WD-induced NAFLD mice. The level of circPI4KB was decreased in the liver tissue and increased in the serum of WD mice compared to those in the control mice (Figure 2D).

### 2.3. circPI4KB Binds to miRNA-122 and Validation of the Circular Structure of circPI4KB

To confirm the ability of circPI4KB to bind miRNA-122, we predicted the binding site of the nucleotide sequence using the ENCORI software (Figure 3A). The pull-down assay with L02 cells was transfected with biotinylated miRNA-122 (50 nM) or biotinylated miRNA-NC and when it was harvested 72 h after the transfection, showed the enrichment of circPI4KB compared with the Biotin-miRNA-NC controls, while circANRIL (the negative control) revealed no enrichment (Figure 3A). Furthermore, after the co-transfection of the reporter vector (pSI-Check2-circPI4KB-wildtype or pSI-Check2-circPI4KB-mutant) and oligonucleotides (miRNA-122 mimics or negative control) in 293T cells, the firefly luciferase activity was measured using a dual-luciferase assay kit against that of the Renilla luciferase. According to the principle of the luciferase reporter assay, if miR-122 binds to the circPI4KB binding site, the luciferase activity will be inhibited. Our results demonstrated that miRNA-122 reduced the luciferase reporter activity by at least 41% compared to the control RNA (Figure 3B). Furthermore, the mutation of the target sites for miRNA-122 revealed no significant difference in the luciferase activity after the transfection of the miRNA-122 into L02 cells (Figure 3B).

CircRNA is another type of RNA with a loop structure without 5′-3′ polarities and polyadenylated tails. Most of the circRNAs are endogenous non-coding RNAs, conserved between different species and showed a higher degree of stability than linear mRNAs. Therefore, it is important to ensure that our circPI4KB are the covalently closed circular structure [25,26]. To validate the circular structure of circPI4KB, we used Sanger sequencing to confirm the head-to-tail splicing (a special splice reaction formed by a 5′-end splice site and the corresponding site at the 3′-end of an exon) in the RT-qPCR product of circPI4KB identified by its expected size and conjunction site (Figure 3C). Convergent primers were designed to amplify *PI4KB* mRNA, and divergent primers were designed to amplify circPI4KB using cDNA and genomic DNA (gDNA). circPI4KB was amplified by divergent primers in cDNA but not in gDNA, which confirmed the circular structure of circPI4KB (Figure 3D). Random hexamer or oligo (dT)18 primers were used in reverse transcription experiments using the RNA from L02 cells. When the oligo (dT)18 primers were used, compared with the random hexamer primers, the relative expression of circPI4KB was significantly downregulated, while that of *PI4KB* mRNA did not change, suggesting that circPI4KB had no poly-A tail (Figure 3E). Moreover, circPI4KB was resistant to RNase R, a highly processive 3′ to 5′ exoribonuclease that digests linear RNAs, confirming that circPI4KB has a circular structure (Figure 3F).

Collectively, these findings demonstrated that circPI4KB could sponge miRNA-122 and circPI4KB is a circular and stable transcript.

### 2.4. The circPI4KB Carried miRNA-122 to Extra-Hepatocyte, Resulting in a Decreased Intra-Hepatic LEVEL and Increased Hepatic Lipid Deposition

Based on the above findings, we hypothesized that circPI4K transports miRNA-122 to extra-hepatocyte, resulting in a decreased level of miRNA-122 in intra-hepatocyte and, therefore, inducing a lipid deposition in hepatocytes. To test this hypothesis, we transfected L02 cells with a circPI4KB overexpression (circPI4KB) or interference (sh-circPI4KB) plasmids, respectively, and then performed FFA inducing. The cells of each group were collected and the amount of TG was detected by an ELISA, which confirmed that the increased expression of circPI4KB aggravates the TG levels in hepatocytes (Figure 4A). The detection of the intracellular lipid deposition by oil red O staining showed that the overexpression of circPI4KB resulted in increased intracellular lipid deposition (Figure 4B). At the same time, the level of miR-122 in hepatocytes was detected by PCR. The results demonstrated that the intra-hepatocyte miRNA-122 was decreased and medium miRNA-122 was increased after the overexpression of circPI4KB; the inhibition of circPI4KB resulted in opposite results (Figure 4C). The PCR and Western blot results suggested that it also altered the levels of the downstream lipid metabolism-related protein and mRNA, including the *FAS, HMGCR*, and *SREBP-2* for cholesterol synthesis and *AGPAT1* and *DGAT1* for TG synthesis (Figure 4D–J).

## 3. Discussion

In the present study, we confirmed that the decreasing level of miRNA-122 in the FFA-induced NAFLD cell model was caused by the excretion of miRNA-122, which surpassed the synthesis. In addition, the level of circPI4KB and miRNA-122 were decreased in intra-hepatocytes but increased in the extra-hepatocyte in the FFA-induced NAFLD cell model, which was consistent with the results in the WD-induced NAFLD mouse model. Furthermore, the binding of circPI4KB and miRNA-122 was confirmed by both RNA pull-down and the luciferase reporter assay. We also showed that the overexpression and downregulation of circPI4KB influenced the distribution of intra- and extra-hepatocyte miRNA-122. The overexpression of circPI4KB decreased the expression level of intra-hepatocyte miRNA-122 and consequently decreased the effect of miRNA-122 in protecting the lipid deposition. Collectively, the present study provides a new explanation for the pathogenesis of the hepatic lipid deposition in NAFLD.

Despite this accumulating evidence on various roles of miRNAs, paradoxical results have been reported related to the disease-associated decrease in the production of intrahepatic of certain miRNAs with an increase in the serum. For example, the inverse correlation has been reported for a miRNA-101 expression in patients with HCC [27] and miRNA-139-5p in patients with primary biliary cirrhosis [28]. Our previous study demonstrated that several miRNAs had an inconsistent or inverse correlation between the circulating and liver tissue expression in patients with NAFLD [14]. In particular, the miRNA-122 expression in the liver tissue decreased 9.27-fold in patients with NAFLD compared to that in the healthy controls and 10.00-fold in patients with NASH compared to that in NAFL. On the contrary, the serum level of miRNA-122 increased 4.31-fold in NAFL vs. the healthy control and 7.28-fold in NASH vs. the healthy control [14]. These phenomena could be explained by: (1) the secretion of miRNA-122 from the hepatocytes to other organs, and (2) how other organs supplied miRNA-122 to the liver during the scanty synthesis of hepatic miRNA-122. Chai et al. experimentally validated that circulating miRNA-122 secreted from the liver as a systemic “hormonal” could enter the muscle and adipose tissues of mice, reducing the mRNA levels of the genes involved in the TG synthesis [13]. In contrast, Baranova et al. proposed a model that the increased secretion of miRNA-122-containing exosomes by adipose tissues increases the supply during the early stages of NAFLD, leading to the reduced intrahepatic production of miRNA-122. However, when the deterioration of adipose catches up with the failing hepatic parenchyma, the external supply of liver-supporting miRNA-122 gradually tapers off, leading to the fibrotic decompensation of the liver and an increase in hepatic carcinogenesis [29]. The miRNA-122 is the most abundant liver-specific miRNA, accounting for approximately 70% of adult liver miRNAs [11]. However, these studies could not explain the alterations in the miRNA-122 levels in the serum of patients with NAFL and NASH, while the liver was in a poor status of lacking protective miRNA-122 [14]. Therefore, we speculated that miRNA-122 could be ‘hijacked’ somehow and hypothesized that circPI4KB carries to extra-hepatocytes and stabilizes it from degradation, leading to an increased serum miRNA-122 in patients with NAFL and NASH.

The exosomes were nano-sized membrane-bound vesicles, serving as novel mediators for long-distance cell–cell communications by transferring various bioactive cargos, such as proteins and RNAs, from their parental cells to distant target cells [30]. In previous studies, miRNAs have been shown to mediate circRNA secretion into exosomes and vice versa. For example, the overexpression of miR-7 in HEK293T cells reduced the level of CDR1as circRNA in exosomes [24]. Moreover, CDR1 as circRNA may also stabilize and transport miRNA-7 in neurons [23]. Li et al. showed that exosome-containing circRNA retained a biological activity as the CDR1as exosomes could abrogate the miR-7-induced growth suppression in receipt cells [24]. The study explained that, at least partly, the exosome-containing circPI4KB and miRNAs-122 bonding complex retains the biological sponging function when arriving at the receipt cells. Taken together, it can be inferred that the mechanism involved in the transportation of miRNA-122 to extra-hepatocyte by circPI4KB could be mediated via exosome.

RNAs are selectively integrated into exosomes via two mechanisms: exo-motif recognition by RBPs, and miRNA-circRNA reciprocal transportation. Regarding the first mechanism involving the exo-motif recognition by RBPs, RNAs are most likely transported into exosomes based on specific exo-motifs in their nucleotide sequence, including the recently proven ‘GGAG/CCCU’ for miRNA, ‘GGAG/CCCU’ for lncRNA, and ‘5ʹ-GMWGVWGRAG-3ʹ’ for circRNA, respectively [31,32,33,34]. We inferred the miRNA-122 losses function of its exo-motif ‘GGAG’ (same site for bonding position) after binding to circPI4KB; thus, the exosome recognized the exo-motif of circPI4KB, consequently “dragging” or “hijacking” the circPI4KB-miRNA-122 binding complex and sorting it into the exosome. However, future studies are required to validate this speculation. Currently, our group is working on the mutation in binding sites and exo-motifs of circPI4KB and miRNA-122, which may provide further insights.

There are also some limitations. First, in our study, circPI4KB was found to inhibit the function of miRNA-122 by adsorbing and transporting miRNA-122 to the extracellular, leading to the lipid deposition in hepatocytes. However, there lacks a validation experiment for the role of miRNA-122 in NAFLD. Nevertheless, the previous several studies have extensively confirmed that the miRNA-122 improves the lipid metabolism by inhibiting triglycerides and cholesterol at a post-transcriptional level [9,12,13]. Second, the functional study of circPI4KB lacks an in vivo experimental verification, which will be further supplemented and deepened in future studies. Finally, we mentioned the possible mechanism of the circRNA adsorption combined with miRNA exocytosis in the discussion, which has not been studied in this study for the time being, but we will do so in future studies, and we will report further when the updated results are available.

## 4. Materials and Methods

### 4.1. Cell and Animal Study

L02 and HEK293T cells from the Cell Bank of Type Culture Collection (Shanghai, China) were cultured in a humidified incubator at 37 °C with 5% CO_2_ using Dulbecco’s Modified Eagle Medium (DMEM; Gibco, Carlsbad, CA, USA) supplemented with 1% penicillin and streptomycin (Invitrogen, Carlsbad, CA, USA) and 10% fetal bovine serum (FBS; Gibco, California). To establish the in vitro model of NAFLD, the cells were cocultured with oleic and palmitic acids (Sigma-Aldrich, St. Louis, MO, USA) at a final concentration of 0.5 mM (FFA, containing oleic acid and palmitic acid at a 2:1 volume ratio) for 24 h [35].

Six to eight-week-old male C57BL/6 mice weighing 18–20 g were obtained from the Animal Experiment Center of Sichuan University. The NAFLD model mice were given a Western diet (WD; 21.1% per kg of fat, 41% sucrose, 1.25% cholesterol) and high-glycemic drinking water (containing 23.1g/L D-fructose, 18.9g/L D-glucose) for 12 weeks, while the control mice were given a normal diet (NC) and ordinary drinking water [36]. The visceral fat and subcutaneous fat in mice were measured using Micro CT (Quantum GX, PerkinElmer, Waltham, MA, USA), scanned in a high-speed mode for 8 sec at a 55 kVp scanning voltage, 5.0 mGy radiation dose, and 38 HU scanning density. The data were analyzed using the Skyscan software 1276 (Micro Photonics, Allentown, PA, USA). All animal experimental procedures were approved by the Animal Care and Use Committee of Sichuan University (Sichuan, China). The experiments were conducted in accordance with the National Research Council’s Guide for the Care and Use of Laboratory Animals (China).

The cell culture medium or mouse serum was collected and tested for the TG. The TG of each group was enzymatically measured (Applygen Technologies Inc., Shanghai, China) against the protein content [37].

### 4.2. Oil Red O Staining

Oil red O staining was used to assess the lipid deposition in the cells or tissues. In the cells, we first performed cell slide experiments on L02 cells. After the cells were stably colonized on the slide, an FFA stimulation (or control treatment) was performed for 24 h, and then the cell slides were collected and fixed for the next oil red O staining. In the tissues, we used frozen sections (10 µm) of the fresh liver for oil red O staining following the procedure. We prepared the oil red staining solution in advance and let it stand for 10 min following the manufacturer’s instructions. The slides with tissue sections were fixed in paraformaldehyde for 10 min and washed with distilled water. Then, they were soaked in 60% isopropyl alcohol for 30 s, followed by oil red O staining solution for 10 min in a dark environment. After dyeing, the slides were rinsed with 60% isopropyl alcohol to remove any excess staining solution and washed with distilled water three times. Finally, the nuclei were stained with hematoxylin for 2 min.

### 4.3. Bioinformatics Analysis

StarBase (https://starbase.sysu.edu.cn, accessed on 19 March 2021), Circbank (http://www.circbank.cn/, accessed on 19 March 2021), and ENCORI (https://starbase.sysu.edu.cn, accessed on 19 March 2021) databases were used to predict which circRNAs could sponge to miRNA-122, and we Venn mapped the main dataset. The circRNAs in the main dataset were separately intersected with liver-specific circRNAs in the CircBase database (http://www.circbase.org/, accessed on 19 March 2021), and the top 20 circRNAs sponging to miRNA-122 were identified. The ENCORI predicted circRNAs were searched in the PubMed GEO dataset of NAFLD mouse circRNA expression profile (GSE134146) to filter the predictive circRNAs that were specifically sponged to miRNA-122.

### 4.4. CircPI4KB Treatment

The overexpression plasmid and shRNA of circPI4KB were synthesized by GenePharma (Shanghai, China), targeting the junction region of the circPI4KB sequence. Except for those without a circPI4KB regulation, the L02 cells were first treated with pcDNA3.1(+)-GFP-CircPI4KB or circPI4KB shRNA plasmid pGPH1/GFP/Neo-sh-circPI4KB for 24 h. The groups without a circPI4KB intervention were treated with a blank plasmid of pcDNA3.1(+)-GFP. All plasmids were transfected with Lipofectamine™ 3000 Reagent (Thermofisher, Waltham, MA, USA) according to the manufacturer’s instructions. Thereafter, free fatty acids were administered to the FFA groups for another 24 h.

### 4.5. Luciferase Reporter Assay

To unravel the circRNA-miRNA interaction, the circPI4KB (circBase: hsa_circ_0006892; circBase, Rajewsky Lab, Berlin, Germany) sequence containing the putative target sites for miRNA-122 was synthesized and cloned into the pSI-Check2 reporter vector (Hanbio Biotechnology, Shanghai, China) downstream to the firefly luciferase (pSI-Check2-circPI4KB-wildtype). The mutant version of circPI4KB (pSI-Check2-circPI4KB-mutant) was also generated with the mutated complementary sites. After the co-transfection of the reporter vector (pSI-Check2-circPI4KB-wildtype or pSI-Check2-circPI4KB-mutant) and oligonucleotides (miRNA-122 mimics or negative control) in 293T cells, the firefly luciferase activity was measured using a dual-luciferase assay kit (Promega, Madison, WI, USA) against that of the Renilla luciferase.

### 4.6. RNA Pull-Down

For a pull-down assay with biotinylated miRNA-122, the L02 cells were transfected with biotinylated miRNA-122 (50 nM) or biotinylated miRNA-NC and harvested 72 h after transfection. The cells were washed with phosphate-buffered solution followed by a brief vortex and incubated in a lysis buffer [20 mM Tris, pH 7.5, 200 mM NaCl, 2.5 mM MgCl_2_, 0.05% Igepal, 60 U/mL Superase-In (Ambion, Austin, TX, USA), 1 mM DTT, protease inhibitors (Roche, Basel, Switzerland)] on ice for 10 min. The lysates were precleared by centrifugation (10,000 rcf, 4min), and 50 μL of the sample was aliquoted for the assay. The remaining lysates were incubated with M-280 streptavidin magnetic beads (ZY130521; Zeye Biotechnology, Shanghai, China). To prevent the non-specific binding of the RNA and protein complexes, the beads were coated with RNase-free bovine serum albumin and yeast tRNA (both from Zeye), and incubated at 4 °C for 3 h. Afterward, the beads were washed twice with ice-cold lysis buffer, three times with the low salt buffer (0.1% SDS, 1% Triton X-100, 2 mM EDTA, 20 mM Tris-HCl (pH 8.0), 150 mM NaCl), and once with the high salt buffer (0.1% SDS, 1% Triton X-100, 2 mM EDTA, 20 mM Tris-HCl (pH 8.0), 500 mM NaCl). The bound RNAs were purified using Trizol for the analysis.

### 4.7. Western Blot Analysis

The samples were lysed in RIPA buffer (CST, Boston, MA, USA) and the proteins were quantified using the bicinchoninic acid method (Thermofisher, Waltham, MA, USA). The lysates were centrifuged (10,000 rcf, 5min), subjected to sodium dodecyl sulfate-polyacrylamide gel electrophoresis, and transferred to polyvinylidene fluoride membranes. Subsequently, the membranes were blocked and incubated with the specific primary antibody overnight at 4 °C, followed by horseradish peroxidase-labeled secondary antibodies (dilution 1:5000–10,000; Abcam, Cambridge, UK). The protein expression was visualized using enhanced chemiluminescence reagents (Amersham, Piscataway, NJ, USA) and the ChemiDoc XRS system (Bio-Rad, Hercules, CA, USA). The primary antibodies used are as follows: 3-hydroxy-3-methyl glutaryl coenzyme A reductases (HMGCR; # sc-271595; Santa Cruz Biotechnology, Santa Cruz, CA, USA), fatty acid synthetase (FAS; #ab133619; Abcam, Cambridge, UK), sterol regulatory element binding protein 2 (SREBP2; #ab30682 Abcam, Cambridge, UK), 1-acyl-sn-glycerol-3-phosphate acyltransferase-beta-1 (Agpat1; #bs-5023R; Bioss, Beijing, China), and diacylglycerol-acyltransferase-1 (Dgat1; #sc-271934; Santa Cruz Biotechnology, Santa Cruz, CA, USA). The antibody for GAPDH (#9485, 1:2500; Abcam, Cambridge, UK) was used as a control.

### 4.8. Real-Time Quantitative PCR Analysis of mRNA, circRNA and miRNA

The total RNAs in the cells or tissues were isolated using a TRIzol reagent (Invitrogen, Carlsbad, CA, USA). The miRNAs were isolated from 200 μL culture media samples using the miRNeasy Mini kit (Qiagen, Redwood City, CA, USA). The concentration of RNA was measured by a Nanodrop (Thermofisher, Waltham, MA, USA), and each paired sample was adjusted to the same concentration. The reverse transcription (RT) of the RNA was performed using the ExScript RT kit (TAKARA, Kusatsu, Japan). RT-qPCR was performed with SYBR Premix Ex Taq (TAKARA, Kusatsu, Japan) and detected by the LightCycler^®^ 96 System (Roche, Basle, Switzerland) according to the manufacturer’s instructions. β-actin and U6 were used as the endogenous controls for mRNA and miRNAs, respectively. The gene expression levels were calculated by the 2^−ΔΔCt^ method. The following primers were used in this study and were designed using the Primer-BLAST tool available at www.ncbi.nlm.nih.gov, accessed on 11 April 2021: circPI4KB (human). Forward primer: CAGCCAGC-AACCCTAAAGTG, reverse primer: ACTGTATCTCCCATGGCCAC; (mouse) forward primer: CTGAAACGAACAGCCAGCAA, reverse primer: GCTCCACTACCATGTCTCC-C. β-actin (human) forward primer: CTCCATCCTGGCCTCGCTGT, reverse primer: GCT-GTCACCTTCACCGTTCC; (mouse) forward primer: CAACTGGGACGACATGGA, reverse primer: CCATCACAATGCCTGTGG. miRNA-122 forward primer: GCCGAGTG-GAGTGTGACAA, reverse primer: GTCGTATCCAGTGCGTGTCG; U6 forward primer: CTCGCTTCGGCAGCACA, reverse primer: AACGCTTCACGAATTTGCGT.

### 4.9. RNase R Treatment

The RNA (2 µg) was treated with RNase R (4 U/μg; Epicentre, Los Angeles, CA, USA) for 15 min at 37 °C or mock-treated. The resulting RNA was purified using the RNeasy MinElute Cleanup Kit (Qiagen, Redwood City, CA, USA). The RNA concentration of the purified samples was determined, and 1 μg of purified RNA was used for the RT.

### 4.10. Statistical Analysis

Student’s *t*-test was performed to analyze two experimental groups and the one-way analysis of variance (ANOVA) was performed to analyze the significant differences among three or more of the experimental groups. The number of samples or experimental replicates was shown in each figure legend. A *p*-value of < 0.05 was considered significant, and the number of “*” represents the degree of significance (* *p* < 0.05; ** *p* < 0.01; *** *p* < 0.001). All statistical analyses were performed in GraphPad Prism 7.

## 5. Conclusions

In conclusion, the present study demonstrates that circPI4KB carries miRNA-122 to extra-hepatocyte, thus decreasing the protective role of miRNA-122 in targeting mRNA and preventing the lipid deposition. The present study provides new insights into the pathogenesis of the hepatic lipid deposition in patients with NAFLD.

## Figures and Tables

**Figure 1 ijms-24-01297-f001:**
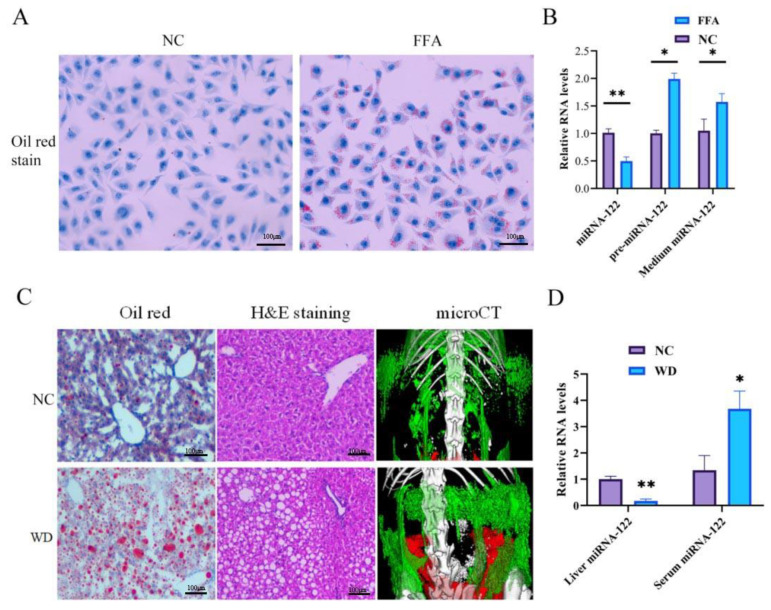
The increased excretion of miRNA-122 from intra-hepatocyte to extra-hepatocyte reduced the level of miRNA-122 in the liver. (**A**) Oil red O staining validated the steatosis in L02 cells without FFA-induced as normal control (NC) and the steatosis in L02 cells with free fatty acid induced (FFA), respectively (40×). (**B**) RT-qPCR analysis of miRNA-122, pre-miRNA-122, and medium miRNA-122 in L02 cells with normal control (NC) or FFA-induced steatosis (FFA), respectively, (*n* = 3). (**C**) The validation of oil red staining, H&E staining, and micro-CT (relative quantitative measurement of abdomen adipose tissue [red] and subcutaneous fat [green]) in Western diet-fed mice (WD) and normal diet-fed mice (NC). (**D**) RT-qPCR analysis of miRNA-122 in liver tissue and serum of WD and control mice, (*n* = 6). (* *p* < 0.05; ** *p* < 0.01).

**Figure 2 ijms-24-01297-f002:**
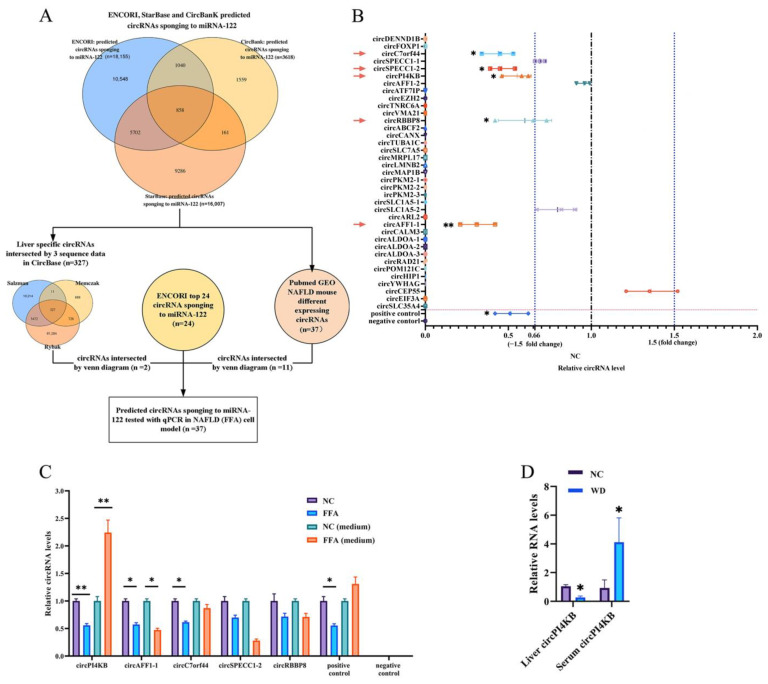
The circPI4KB was decreased in hepatocytes and was increased in extra-hepatocyte. (**A**) Flowchart of prediction of specific circRNAs sponging to miRNA-122. The ENCORI, StarBase, and Circbank databases were used to predict the circRNAs sponging to miRNA-122. The predictive circRNAs were further intersected with 1) liver-specific circRNAs (down left) and GEO NAFLD mouse profile (down right) (Circbase of Memczak mouse 1903 circRNA intersected with 1686 differentially expressed circRNA in GSE134146 dataset of NAFLD model). Finally, 37 predicted circRNAs were identified that sponge to miRNA-122 and were predicted to be possibly dysregulated in NAFLD. (**B**) RT-qPCR analysis of the 37 predicted miRNA-122-sponging circRNAs were validated in the FFA-induced NAFLD cell model. (**C**) RT-qPCR analysis of the expression level of 5 dysregulated circRNAs in normal control cell (NC cell) and culture medium (NC medium), and FFA-induced NAFLD cell (FFA cell) and culture medium (FFA medium), (*n* = 3). (**D**) RT-qPCR analysis of the expression level of circPI4KB in liver tissue and serum of WD-induced NAFLD (WD) and control mice (NC), (*n* = 6). (* *p* < 0.05; ** *p* < 0.01).

**Figure 3 ijms-24-01297-f003:**
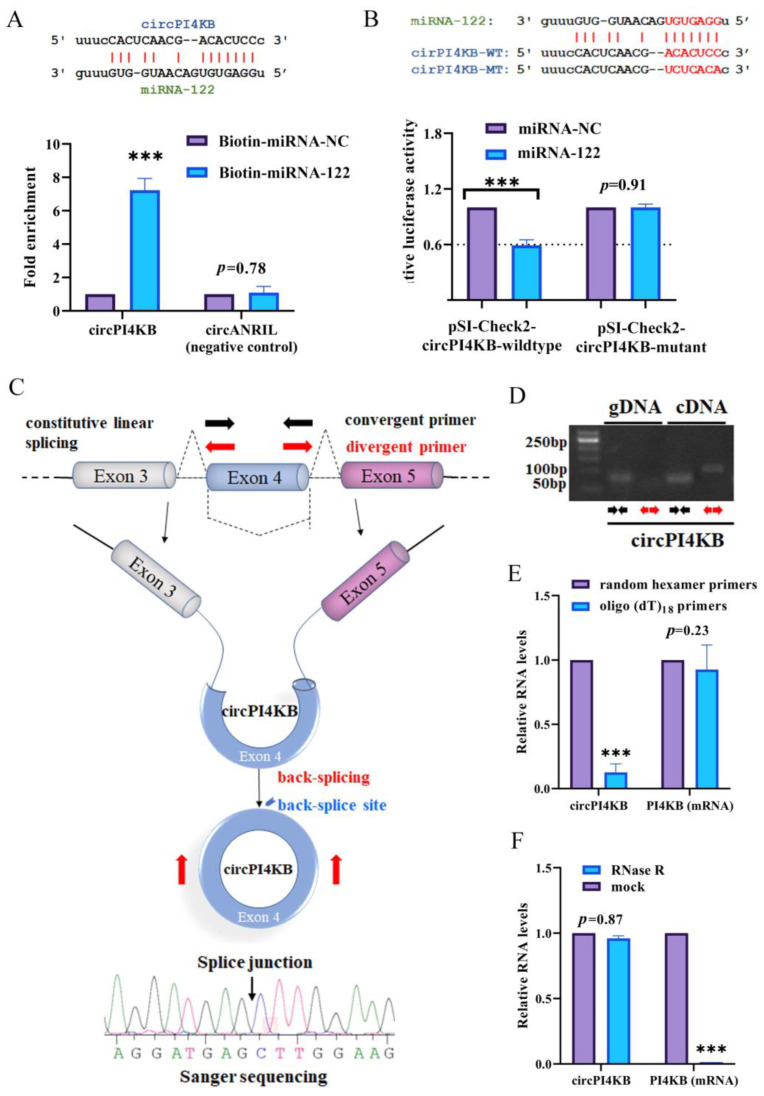
The validation of binding of circPI4KB and miRNA-122 and the circular structure of PI4KB. (**A**) RT-qPCR showed the level of circPI4KB in the streptavidin-captured fractions from the L02 cell lysates after transfection with 3′-end biotinylated miRNA-122 or biotinylated miRNA-NC, while circANRIL was known to not bind to miRNA-122 as negative control, (*n* = 3). (**B**) The luciferase activity of wild-type plasmid of luci-circPI4KB-wildtype (pSI-Check2-circPI4KB-wildtype) or mutation plasmid of luci-circPI4KB-mutant (pSI-Check2-circPI4KB-mutant) in L02 cells after co-transfection with miRNA-122, (*n* = 3). (**C**) Scheme illustrating the PI4KB exons 4 circularizations to form circPI4KB. The specific PCR primers used to detect circPI4KB by RT-qPCR are indicated by red arrows below. The amplified product of specific divergent primers was confirmed according to the sequence of circPI4KB by sequencing. The black arrow represents head-to-tail circPI4KB splicing sites. (**D**) Convergent primers designed to amplify *PI4KB* mRNA, and divergent primers designed to amplify circPI4KB using cDNA and genomic DNA (gDNA). (**E**) Divergent primers amplify circPI4KB from cDNA but not genomic DNA. Random hexamer or oligo (dT)18 primers were used in the reverse transcription experiments. The relative RNA levels were analyzed by RT-qPCR and normalized to the value using random hexamer primers, (*n* = 3). (**F**) The relative RNA levels analyzed by RT-qPCR and normalized to the value detected in the mock group after RNase R treatment, (*n* = 3). (*** *p* < 0.001).

**Figure 4 ijms-24-01297-f004:**
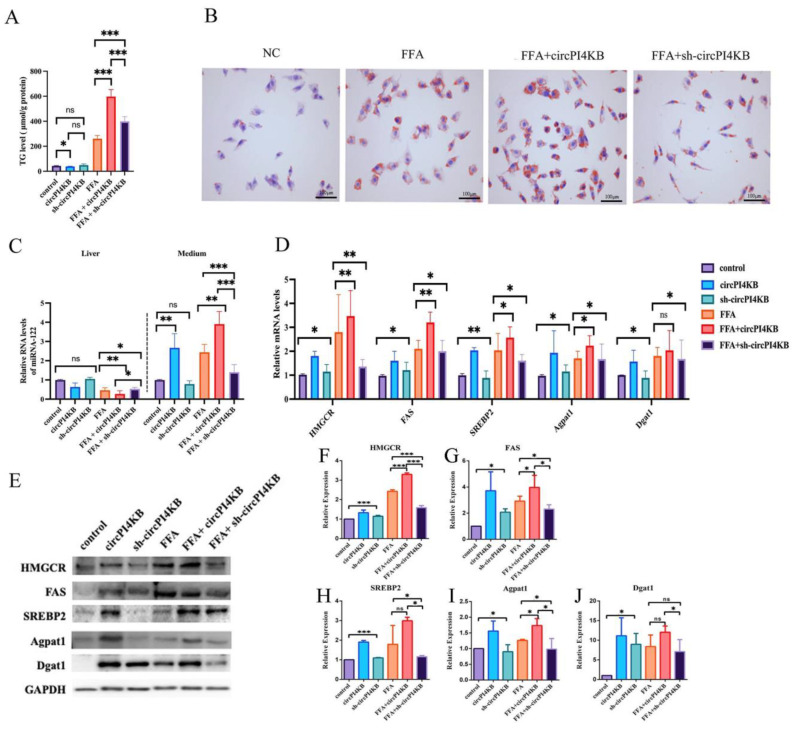
circPI4KB transported the protective miRNA-122 to extra-hepatocyte, resulting in lipid deposition in hepatocyte due to decreased intra-hepatocyte level of miRNA-122, (*n* = 3). (**A**) Enzyme-linked immunosorbent assay was used to detect the level of TG in each group. (**B**) Oil red O staining validated the steatosis in L02 cells with FFA-induced and overexpressed (circPI4KB) or inhibited plasmids (sh-circPI4KB) transfected, respectively (40×). (**C**) RT-qPCR showed the level of expression of intra- and extra-hepatocyte miRNA-122 after transfection with overexpressed (circPI4KB) or inhibited plasmids (sh-circPI4KB). (**D**) RT-qPCR analyses showed the modification of mRNA and protein levels of miRNA-122-downstream cholesterol synthesis (*HMGCR*, *FAS*, *SREBP2*) and TG synthesis (*Agpat1* and *Dgat1*) after overexpressing or inhibiting circPI4KB. (**E**–**J**) Western blot (**E**) and gray value analysis (**F**–**J**) showed the protein levels of HMGCR, FAS, SREBP2, Agpat1, and Dgat1 after overexpressing or inhibiting circPI4KB. (* *p* < 0.05; ** *p* < 0.01; *** *p* < 0.001; ns: not significant).

## Data Availability

The data presented in this study are available in the article.

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
