# Peer review of "CircRNA-PI4KB Induces Hepatic Lipid Deposition in Non-Alcoholic Fatty Liver Disease by Transporting miRNA-122 to Extra-Hepatocytes"

_ijms, 2023, doi:10.3390/ijms24021297_

Round 1
Reviewer 1 Report
This study showed that miR-122 localization was affected during non-alcoholic fatty liver disease both in vitro and in vivo. This study also showed that circRNA-PI4KB was increased in non-alcoholic fatty liver disease and bound to miR-122 to transport it outside of hepatocytes. This study is interesting and introduces a novel pathogenic pathway of non-alcoholic fatty liver disease. That being said, there were some issues that need to be addressed.
#1 FFA was never defined or explained before it was introduced in the Results section.
#2 The Results section as a whole needs more detailed explanations of the experimental procedures. These explanations do not need to be as detailed as the Methods section but should be helpful to follow along with the results shown.
#3 Figure 1A should be explained better to justify why it is being shown.
#4 The Western diet mouse model should be introduced and explained better in the Results section.
#5 Figure 1C should also be explained to justify why it is being shown.
#6 In the Results, there was only one sentence that mentioned establishing a mouse model. However, this mouse model was only used for two qRT-PCR experiments. There needs to be a better explanation on why an entire mouse model was established. For example, a phrase like “…to further assess miR-122 in an in vivo model…” is all that is needed.
#7 On line 103-104, “as the main dataset” is out of place. What exactly was the main dataset used here.
#8 On line 110-111, the phrase “with the same tendency as miR-122” is not shown in Figure 1C. Therefore, this phrase should be explained further. Where do we see a similar trend in circRNAs and miR-122 expression?
#9 In Section 2.3, a more detailed explanation of the Luciferase reporter assay is needed.
#10 What was the importance of validating the circular structure of circRNA-PI4KB? A simple sentence explaining the justification of this experiment is all that is needed.
#11 In Section 2.4, a more detailed explanation of the experiment explaining the overexpression and the shRNA silencing is needed.
#12 In Section 4.2, the Oil Red O staining only describes the staining on animal slides. However, Figure 1A depicts L02 cells. Therefore, an additional explanation of the Oil Red O staining on these cells is needed.
#13 In Section 4.5, a Luciferase assay was conducted on HEK293T cells. However, there is no information on how these cells were cultured, maintained, and/or treated.
#14 Have the authors knocked down miR-122 expression in any of these cell lines? The addition of a miR-122 knockdown to Figure 4C would show that miR-122 influences these proteins’ expression.
#15 On line 407, it is stated that circRNA-PI4KB decreases the protective role of miR-122 in preventing lipid depositions. However, this is not currently shown in any of the figures.
#16 Have the authors compared Oil Red O staining for lipid steatosis in FFA-induced L02 cells, FFA-induced L02 cells with overexpressed circRNA-PI4KB, FFA-induced L02 cells with shRNA knockdown of circRNA-PI4KB, and L02 cells with a knockdown of miR-122? This experiment would explain why it was important to include Figure 1A and would allow you to conclude that circRNA-PI4KB does decrease the protective role of miR-122 in preventing lipid depositions.
Author Response
International Journal of Molecular Sciences
Re: Manuscript reference ijms-2067097
Dear reviewer:
Thank you very much for giving us the opportunity to revise our article, titled “CircRNA-PI4KB induces hepatic lipid deposition in non-alcoholic fatty liver disease by transporting miRNA-122 to extra-hepatocytes” (revise ijms-2067097). We thank you for the time and effort to review the manuscript. Your suggestions have enabled us to improve our work. We revised our paper as suggested. Accordingly, we have uploaded a copy of the revised manuscript with all the changes highlighted for easy check/editing purpose.
In the following pages are our point-by-point responses to each of the comments. Revisions in the text are using the “Track Changes” function for changes and additions. We hope that the revisions in the manuscript and our accompanying responses will be sufficient to make our manuscript more suitable for publication.
Please see the attachment.
We look forward to hearing from you regarding our submission. We would be glad to respond to any further questions and comments that you may have.
Sincerely,
Hong Tang
Center of Infectious Diseases,
West China Hospital of Sichuan University
E-mail: [email protected]

Reviewer 2 Report
miRNA-122, the most abundant liver-specific miRNA, protects hepatic steatosis and inhibits cholesterol and fatty acid synthesis in NAFLD.
The author showed with in vitro and in vivo experiment, that miRNA-122 decreased in hepatocytes or liver tissue and increase in extra-hepatocytes or serum. They suggested that miR-122 escaped from intra- to extra-hepatocytes due to excretion.
Using bioinformatics tools, they showed that miRNA-122 binds to circPI4KB. The validation of these results by RNA pull-down and luciferase reporter assay showed that the levels of circPI4KB in intra- and extra-hepatocytes corresponded to that of miRNA-122, and the overexpression of circPI4KB increased the miRNA-122 in extra-hepatocytes, consequently accomplishing a decreased protective role of miRNA-122 in inhibiting lipid deposition.
The author provided an interesting explanation for the pathogenesis of hepatic lipid deposition in NAFLD.
The topic has been investigated extensively and they demonstrated that circPI4KB carries miRNA-122 to extra-hepatocyte, thus decreasing the protective role of miRNA-122 in targeting mRNA and preventing lipid deposition.
Minor revision
- The introduction still lacks various information that would make reading and understanding the manuscript easier. Information on circRNA is limited. It is necessary to be describe how the circRNA works with miRNA together for transport. In addition, the most important studies in this area (from line 206) should be mentioned in the introduction instead of the discussion. Information on the most important miRNAs associated with hepatic diseases is also missing in the introduction.
- Figure 2 : the graphics are very small and the legends are not readable
- The sample numbers and the repetitions for the different experiment should be described in the figures or in methods and materials
- Figure 4 (C): From the Western-blot it is difficult to determine the expression level of downstream lipid-metabolism-related proteins. A diagram is necessary to be done.
Author Response

(The authors gave the same response as above.)

Round 2
Reviewer 1 Report
Thank you for addressing my peer-review comments. The revisions and additions you have made satisfy my concerns, and I recommend that the manuscript be published after English editing for grammar and syntax.